# Assessment of Decongestion Status Before Discharge in Acute Decompensated Heart Failure: A Review of Clinical, Biochemical, and Imaging Tools and Their Impact on Management Decisions

**DOI:** 10.3390/medicina61050816

**Published:** 2025-04-28

**Authors:** Diana-Ligia Pena, Adriana-Mihaela Ilieșiu, Justin Aurelian, Mihai Grigore, Andreea-Simona Hodorogea, Ana Ciobanu, Emma Weiss, Elisabeta Badilă, Ana-Maria Balahura

**Affiliations:** 1Department of Cardiology, “Prof. Dr. Theodor Burghele” Clinical Hospital, 061344 Bucharest, Romania; dianaligiapena@gmail.com; 2Department of Cardiology, “Prof. Dr. Theodor Burghele” Clinical Hospital, Carol Davila University of Medicine and Pharmacy, 050474 Bucharest, Romania; mihai.grigore@umfcd.ro (M.G.); andreea.hodorogea@umfcd.ro (A.-S.H.); ana-maria.balahura@umfcd.ro (A.-M.B.); 3Department of Urology, “Prof. Dr. Theodor Burghele” Clinical Hospital, Carol Davila University of Medicine and Pharmacy, 050474 Bucharest, Romania; justin.aurelian@umfcd.ro; 4Department of Cardiology, “Prof. Dr. Agrippa Ionescu” Emergency Clinical Hospital, Carol Davila University of Medicine and Pharmacy, 050474 Bucharest, Romania; ana.ciobanu@umfcd.ro; 5Department of Cardiology, Colentina Hospital, Carol Davila University of Medicine and Pharmacy, 050474 Bucharest, Romania; emma.weiss@umfcd.ro (E.W.); elisabeta.badila@umfcd.ro (E.B.)

**Keywords:** acute decompensated heart failure, congestion assessment, decongestion strategies, multimodal evaluation tools, clinical decision-making, discharge planning, prognostic biomarkers, imaging techniques

## Abstract

Acute decompensated heart failure (ADHF) represents a major healthcare burden, with residual congestion at discharge being a critical determinant of poor outcomes. Despite its prognostic significance, the assessment of decongestion status before discharge remains suboptimal, highlighting the need for a more comprehensive evaluation approach. This descriptive review synthesizes current evidence on congestion assessment methods in ADHF, focusing on their role in discharge decision-making and prognostic value. We describe various evaluation tools, including clinical examination, biomarkers, imaging techniques, and congestion scores, presenting their integration into a practical assessment algorithm. A comprehensive algorithm for congestion assessment before discharge is presented, incorporating multimodal evaluation techniques, with the aim of highlighting the practical utility of various assessment methods in guiding treatment decisions and determining optimal discharge timing. Integration of multiple parameters provides superior accuracy in evaluating decongestion status compared to single-method approaches. A standardized, multimodal approach to congestion assessment before discharge is essential for optimal ADHF management. The proposed assessment algorithm, combining clinical, biochemical, and imaging parameters, offers a practical framework for more reliable discharge decision-making, potentially improving patient outcomes.

## 1. Introduction

Acute decompensated heart failure (ADHF) represents one of the leading causes of hospitalization worldwide, and it is associated with significant morbidity, mortality, and healthcare costs. Despite advances in therapeutic strategies, approximately 25% of patients are readmitted within 30 days of discharge, with residual congestion being identified as a major contributing factor to poor outcomes [1,2,3,4].

The presence and persistence of congestion remain central to the pathophysiology and clinical manifestations of heart failure (HF). Studies have consistently demonstrated that residual congestion at discharge is strongly associated with increased mortality and rehospitalization rates. This relationship highlights the critical importance of achieving optimal decongestion before discharge, yet current clinical practices often fall short of this goal [5,6].

The pathophysiology of HF symptoms involves two distinct congestion phenotypes:intravascular congestion (increased fluid volume within the vascular system) which manifests as jugular venous distension (JVD) and hepatojugular reflux, directly reflecting elevated central venous pressure (CVP > 10 mmHg), and also as dyspnea, orthopnea, and bendopnea [7];tissue congestion (extravascular congestion) which presents as peripheral edema, pulmonary rales, and ascites, indicating fluid extravasation into interstitial spaces due to sustained capillary leakage [8].

Traditional clinical assessment of congestion status presents significant challenges. Physical examination findings alone have shown a poor correlation with actual hemodynamic status, with studies revealing substantial discordance between clinical assessment and invasive hemodynamic measurements [9,10,11].

This limitation has stimulated the development and validation of various complementary assessment tools, including biomarkers, imaging techniques, and composite congestion scores [12].

Recent evidence suggests that a multimodal approach to congestion assessment may provide more reliable information for clinical decision-making. The integration of newer technologies, such as lung ultrasound [13], renal ultrasound [14], bioimpedance analysis [15], and point-of-care biomarker testing, alongside traditional clinical evaluation, have shown promise in better identifying patients at risk for poor outcomes [1].

Some studies strongly recommend the use of natriuretic peptides in guiding discharge management, which has demonstrated potential benefits in optimizing treatment strategies, particularly in reducing mortality and readmission rates [12]. Studies have shown that point-of-care ultrasound assessment significantly improves congestion evaluation and clinical decision-making [16].

The timing and trajectory of decongestion during hospitalization have emerged as crucial factors affecting outcomes. Early hemoconcentration, occurring within the first 48 h of hospitalization, is associated with the lowest occurrence of mortality and HF rehospitalization [17,18].

Trials such as EVEREST (Efficacy of Vasopressin Antagonism in Heart Failure: Outcome Study with Tolvaptan), a multicenter trial with 4133 patients enrolled from over 300 sites from North America, South America, and Europe, demonstrated that despite substantial improvement in congestion with standard therapy, patients with minimal or absent congestion signs at discharge still experienced high mortality and readmission rates [19].

Recent data from RELAX-AHF-2 (RELAXin in Acute Heart Failure-2), another large-scale, multicenter trial, with approximately 6600 patients enrolled, showed that residual congestion at day 5 was present in 57% of patients and independently associated with worse outcomes [20].

The optimal monitoring approach remains challenging, as clinical assessment alone proves insufficient. Studies have shown that more than one-third of patients (35%) had persistent moderate to severe congestion at discharge despite aggressive inpatient therapy [5].

Despite these advances, there remains no standardized approach to evaluating congestion status before discharge, and the optimal combination and timing of assessment methods continue to be subjects of investigation. This review aims to synthesize current evidence on available congestion assessment methods in ADHF (Appendix B, Table A1, Table A2 and Table A3), evaluate their practical application in clinical decision-making, and propose an integrated approach to pre-discharge congestion assessment.

Through this review, we aim to provide clinicians with a practical framework for optimizing discharge decisions and improving patient outcomes in ADHF.

## 2. Methods of Congestion Assessment and Their Role in Discharge Decision-Making

### 2.1. Methods of Congestion Assessment

#### 2.1.1. Clinical Evaluation

Traditional clinical assessment of congestion in HF remains challenging due to variable correlations between symptoms, physical findings, and true hemodynamic status [21] (Appendix B, Table A1).

Dyspnea remains the cardinal symptom of pulmonary congestion. However, dyspnea has limited specificity (52%) and sensitivity (66%) for HF, as it overlaps with respiratory and systemic conditions [5].

Improvement of this symptom can be a primary goal of ADHF treatment and can be one of the indicators of the patient’s discharge readiness. Its resolution, alongside with improvement in NYHA (New York Heart Association) functional class, should be key considerations in discharge decision-making. However, dyspnea’s resolution is not sufficient to determine complete decongestion. The EVEREST trial demonstrated that 35% of patients with resolved dyspnea still had significant residual congestion at discharge [5].

Orthopnea and paroxysmal nocturnal dyspnea (PND) show a stronger association with elevated left ventricular filling pressures (LVFP > 25 mmHg), often manifesting late in congestion progression, making their resolution a more reliable indicator of successful decongestion. These symptoms’ persistence suggests incomplete decongestion and the need for further treatment and daily reassessing [22]. Regardless, their absence does not exclude subclinical hemodynamic congestion [23].

The analysis of the EPICA registry from Portugal identified that the descriptors of dyspnea at rest, orthopnea, and previous PND were highly specific (99%) for HF; nonetheless, the sensitivity of these symptoms was relatively low [24]. This indicates that while the presence of orthopnea and PND is strongly indicative of HF, their absence does not necessarily rule out the condition or indicate readiness for discharge.

Jugular venous distention (JVD) is an important clinical sign for assessing right-sided filling pressures and overall volume status, but studies show only 42% of decompensated patients exhibit JVD [13], limiting its utility as a sole indicator of discharge readiness. In the evaluation of JVD at the supraclavicular point, a study reported that when using the traditional method, the sensitivity was 70%, specificity 76%, positive predictive value (PPV) 59%, and negative predictive value (NPV) 91%. In addition, when the measurement of right atrial depth was corrected for error, sensitivity improved to 90%, and the NPV reached 96% [25].

Pulmonary rales, or crackles, are a critical clinical sign in ADHF, often indicating pulmonary congestion due to elevated LVFP and fluid accumulation in the alveoli. Their presence correlates with the severity of pulmonary congestion and can serve as a prognostic marker. Studies demonstrate that while 86% of acute decompensation cases present with pulmonary crackles, these auscultatory findings often emerge late in congestion progression, correlating with advanced interstitial edema [23,26].

Patients with rales over more than two-thirds of their lung fields upon admission have been shown to be at higher risk of adverse outcomes, including increased mortality and rehospitalization rates. However, the sensitivity of rales for diagnosing ADHF varies widely, ranging from 13% to 70%, with specificity between 35% and 100%, depending on the population studied and the criteria used for detection. Pulmonary crackles are present in 86% of acute decompensations but absent in early congestion and often lack in chronic pulmonary venous congestion despite elevated filling pressures [13].

Resolution of or significant reduction in rales suggests improved pulmonary congestion but not complete resolution, while persistent rales may indicate incomplete decongestion or the need for further treatment.

Pleural effusions (PEs) are a frequent finding in patients with ADHF, occurring in approximately 46–50% of cases [27]. The accurate identification of PEs as transudates in ADHF remains challenging. Clinical judgement alone has demonstrated low sensitivity, with misclassification rates of 44–48% [28]. PEs are predominantly bilateral (58%) or right-sided (27%), with left-sided effusions being less common (14%). PEs are more prevalent in males and are associated with elevated systolic pulmonary artery pressure (sPAP) and high serum levels of NT-proBNP [27].

PEs can indicate residual hemodynamic congestion, which may not be clinically apparent. Persistent congestion at discharge is associated with increased risk of rehospitalization and poor outcomes [29]. The presence of PEs should be considered when evaluating discharge readiness. Noninvasive assessments, such as ultrasound evaluation of the inferior vena cava and bioimpedance measures, can help estimate hemodynamic status [29]. The ongoing TAP-IT trial (Thoracentesis to Alleviate cardiac Pleural effusion Interventional Trial) is investigating the effect of up-front thoracentesis in addition to standard therapy for ADHF patients with PEs [30]. This randomized controlled trial aims to assess if this approach can increase days alive and out of hospital in the 90 days following randomization [30].

Another late manifestation of fluid overload, peripheral edema, typically appears after 3.5–7 kg of weight gain. Peripheral edema is a significant clinical sign in ADHF, reflecting fluid retention due to impaired cardiac function and neurohormonal activation. It often results from elevated CVP, leading to fluid accumulation in the interstitial tissues, particularly in the lower extremities. However, peripheral edema can be a manifestation of various underlying conditions, including adverse reaction to diverse medication, venous insufficiency, nephrotic syndrome, lymphedema, and others. That suggests that while the presence of this sign may be suggestive of a cardiac cause, its absence does not rule out other etiologies [31].

When it is present, peripheral edema is not only a marker of disease severity but also a predictor of adverse outcomes. For instance, studies have shown that residual congestion, including peripheral edema, at discharge is associated with increased mortality and rehospitalization rates [32].

Hepatomegaly is a common physical finding in patients with ADHF, serving as an important indicator of right-sided heart dysfunction and systemic congestion, and it results from increased CVP and hepatic congestion [33]. In a large case series of 175 patients with acute and chronic HF, hepatomegaly was present in 90% to 95% of cases, highlighting its prevalence in this patient population [34].

The persistence of hepatomegaly at the time of discharge may indicate residual congestion, which is associated with an increased risk of readmission and poor outcomes. The ESCAPE trial (Evaluation Study of Congestive Heart Failure and Pulmonary Artery Catheterization Effectiveness), a large-scale study with 433 patients enrolled, revealed that hepatomegaly was frequently observed, alongside a high prevalence of liver function test abnormalities at baseline, with 46% of patients showing elevated aspartate aminotransferase (AST) levels, and persistent liver function abnormalities during hospitalization were linked to poorer outcomes, including increased mortality and rehospitalization rates [35].

Weight gain precedes visible edema by 3.5–7 kg, making daily measurements all-important. A 2.3% body weight increase over 3 days (∼1.8 kg for 80 kg patient) predicts impending decompensation with 89% accuracy [36].

Pre-hospitalization fluid retention in HF patients can vary, sometimes reaching 10–15 kg. During treatment, clinicians typically aim for a negative fluid balance of 1 L (or 1 kg) per day through diuresis. Clinical trials of hospitalized HF patients with congestion generally report an average weight loss of 3–4 kg by discharge [6,37,38]. Net negative fluid balance during hospitalization is associated with improved outcomes, but inaccuracies in intake/output recording limit utility.

Daily weight measurements, accurate fluid balance charting along with monitoring diuresis in ADHF are crucial for effective management, urine output monitoring being especially essential for assessing diuretic response. The DOSE study emphasized the importance of assessing loop diuretic response through multiple measures, including dyspnea relief, weight change, net fluid loss, and NT-proBNP level changes [37].

While the term “dry weight” is more commonly used in the context of dialysis, the concept is relevant to HF management because it represents the optimal fluid status for a patient with HF: the patient’s ideal weight without any excess fluid retention, similar to its use in dialysis patients. In ADHF, targeting a “dry weight” has been a cornerstone strategy for measuring congestion relief. However, studies have shown that despite aggressive diuresis, 35–40% of patients were still moderately congested at discharge, indicating that achieving the ideal dry weight can be challenging [39]. Emerging technologies such as relative plasma volume monitoring and body impedance analysis may help in assessing dry weight in the future [40].

However, despite ESC guidelines mandating strict intake/output (I/O) tracking, audit data reveal that only 52% of acute HF admissions had documented fluid balance charts and charts were completed accurately in merely 29% of hospitalization days [36].

Ascites, the accumulation of fluid in the peritoneal cavity, is an important clinical finding in patients with ADHF. While less common than other signs of congestion, ascites carries significant implications for patient management and outcomes. Ascites occurs in approximately 20–50% of patients with advanced HF [41], and this finding often indicates more severe disease and significant fluid overload.

Persistent ascites at discharge is a red flag indicating incomplete decongestion and increased risk of poor outcomes. The ESCAPE trial demonstrated that residual congestion, including ascites, was associated with higher rates of rehospitalization and mortality in ADHF patients [42].

Future research should focus on developing standardized protocols for assessing and managing ascites in ADHF to improve patient outcomes and reduce readmission rates. Additionally, incorporating advanced monitoring techniques, such as point-of-care ultrasound for ascites assessment, may provide more precise quantification and guide decongestion strategies more effectively [42].

In conclusion, physical examination findings alone have shown a poor correlation with actual hemodynamic status, with studies revealing substantial discordance between clinical assessment and invasive hemodynamic measurements [20].

While these symptoms and signs are crucial in assessing readiness for discharge, no single parameter should be used in isolation. A comprehensive evaluation of multiple clinical indicators should guide the discharge decision-making process. Standardized protocols incorporating these elements can help ensure consistent, evidence-based discharge practices and improve outcomes for ADHF patients.

#### 2.1.2. Laboratory Assessment

1.Natriuretic Peptides (NPs)

Natriuretic peptides—B-type natriuretic peptide (BNP) and the N-terminal fragment of proBNP (NT-proBNP)—represent essential biomarkers in HF, playing crucial roles in diagnosis and prognostics [43] (Appendix B, Table A2). The Breathing Not Properly study published in 2002 demonstrated that BNP measurements significantly improved ADHF diagnosis in patients with acute dyspnea with an unclear diagnosis of AHF [44,45].

Jumping just three years ahead, in 2005, the Pro-BNP Investigation of Dyspnea in the Emergency Department (PRIDE) study concluded that NT-proBNP could also assist with diagnosing ADHF in patients with acute dyspnea of an unknown cause [46].

Moreover, the prognostic significance of NPs is important at discharge, with studies showing that discharge NT-proBNP levels predict outcomes similarly in both heart failure with preserved ejection fraction (HFpEF) and heart failure with reduced ejection fraction (HFrEF) populations, with a hazard ratio for mortality of 2.14 and 1.96, respectively, for every 2.7-factor increase in NT-proBNP levels [47].

Multiple studies, including data from the OPTIMIZE-HF registry with over 7000 patients, have demonstrated that discharge NP levels hold more prognostic value for mortality and HF readmission than their admission levels. A reduction of ≥30% in NT-proBNP from admission to discharge identifies lower-risk patients with better outcomes [48,49].

These findings demonstrate that the timing of discharge could be guided by the natriuretic peptide pattern, as discharge values have superior prognostic value compared to admission levels. The burden of ADHF remains significantly high, with up to 40.1% of patients experiencing readmission or death within a year of discharge, according to the European Society of Cardiology (ESC), which emphasizes the importance of appropriate discharge timing, taking into account this biomarker guidance [50].

Current evidence supports a two-measurement approach: initial assessment at admission for diagnosis and risk stratification, followed by pre-discharge measurement to evaluate treatment response and guide discharge timing. This strategy allows for more precise risk stratification and optimization of the discharge decision, which is particularly important given that discharge BNP levels below 250–350 pg/mL are associated with the lowest risk for mortality and HF readmission [51].

An important mention is that Point-of-Care Testing (POCT) has significantly reduced the time required for obtaining natriuretic peptide (NP) results after a patient’s arrival at the hospital. POCT can be performed and analyzed in close proximity to the patient, typically yielding results within 20–30 min. This rapid turnaround time enables healthcare providers to make timely diagnostic and treatment decisions for patients with suspected HF [51]. At this moment, NP levels can be determined using POCT just as accurately as laboratory platforms, which makes it excellent for serial testing, even before discharge [52].

Several limitations need to be mentioned for the use of NPs. Atrial fibrillation significantly reduces the specificity and accuracy of NPs in diagnosing ADHF, with the area under the curve dropping from about 0.9 in sinus rhythm to approximately 0.7 in atrial fibrillation [53]. Obesity can lead to unexpectedly lower NP concentrations, potentially causing false negatives in 15–20% of ADHF cases [54]. Age is an independent determinant of NP plasma concentrations, necessitating age-adjusted values for enhanced specificity [53].

2.Hemoconcentration

Hemoconcentration is a useful parameter in evaluating decongestion and making discharge decisions for patients with ADHF, defined as an increase in the concentration of red blood cells and plasma proteins in the blood, indicating a reduction in intravascular volume. In the context of ADHF treatment, hemoconcentration occurs when the rate of fluid removal exceeds the rate of plasma refilling, suggesting effective decongestion [55].

Evaluation of hemoconcentration can be performed using changes in commonly measured parameters such as hemoglobin, hematocrit, serum albumin, or total protein [56]. While no prospective trials have specifically examined hemoconcentration as a target for guiding diuretic therapy in ADHF patients, its correlation with improved outcomes has been consistently observed. The rise and fall of hemoconcentration values should be interpreted in the context of the overall clinical picture: a rising hemoconcentration generally indicates effective fluid removal exceeding the plasma refill rate, and a falling hemoconcentration may suggest either inadequate decongestion or equilibration of extravascular fluid with the intravascular space [18].

A retrospective analysis of the PROTECT trial revealed that anemia was present in approximately half of ADHF patients, with 69% experiencing an increase in hemoglobin levels during hospitalization. This analysis found that hemoconcentration was independently associated with reduced 180-day mortality (HR 0.66; 95% CI, 0.51–0.86; *p* = 0.002), despite more frequent occurrences of worsening renal function during decongestion therapy. Notably, initial hemoglobin levels were not predictive of outcomes [56].

In a study by Darawsha et al., 704 ADHF patients with volume overload were assessed for changes in congestion scores and hemoconcentration, defined as increases in both hemoglobin and hematocrit between admission and discharge. The results showed that hemoconcentration was linked to improved survival over a mean follow-up of 14 months (adjusted HR 0.70; 95% CI, 0.54–0.90; *p* = 0.006). Conversely, persistent congestion at discharge was associated with poorer survival. However, the correlation between hemoconcentration and clinical assessment of decongestion was found to be weak [57].

Hemoconcentration serves as a proxy for intravascular volume reduction. During aggressive diuresis, intravascular volume depletion surpasses plasma refill from the extravascular space, but this does not necessarily indicate that the patient has achieved euvolemia. Research by Testani et al. demonstrated that early decongestion, likely indicative of volume contraction without excessive extravascular volume depletion, did not confer a mortality benefit. However, late hemoconcentration, more likely associated with sustained decongestion and euvolemia, predicted improved survival (HR, 0.74; 95% CI, 0.59–0.93; *p* = 0.009). Late hemoconcentration was also correlated with greater weight loss, higher cumulative diuretic doses, and shorter hospital stays [18].

In the context of discharge decision-making, hemoconcentration plays several important roles. The presence of hemoconcentration, especially when late and sustained, may indicate adequate decongestion and guide the discharge decision. Patients exhibiting hemoconcentration have a lower risk of adverse post-discharge events, which can influence follow-up planning and post-hospitalization management.

While hemoconcentration has shown promise in identifying patients who have undergone aggressive decongestion and has been associated with improved post-discharge survival, it is essential to recognize its limitations and potential pitfalls in clinical decision-making. Hemoconcentration in isolation does not provide information about total body volume status, as it merely indicates that fluid has been removed from the intravascular space faster than it could be replaced by extravascular fluid. Acute hemorrhage leads to a decrease in blood volume, which can mask the hemoconcentration effect of diuresis. Fluid perfusions can dilute the blood, counteracting the hemoconcentration effect of diuresis and potentially masking effective decongestion, and the list can continue, but the main point remains that it is important to recognize its limitations.

3.Renal function markers

Renal function markers play an important role in assessing congestion status and optimizing discharge timing in ADHF.

Estimated glomerular filtration rate (eGFR) is an independent predictor of mortality in HF patients, with each 10 mL/min decrease in baseline eGFR associated with a 1.064 increased risk of death. Patients with eGFR below 60 mL/min demonstrate significantly higher mortality risk, with hazard ratios of 1.32 for eGFR 30–59 mL/min and 2.54 for eGFR 15–29 mL/min [58]. At admission, approximately 53% of patients present with moderately or severely reduced eGFR (<60.0 mL/min/1.73 m^2^) [59].

Worsening renal function (WRF) is defined as an increase during the hospital stay in serum creatinine ≥0.3 mg/dL or ≥125% of baseline (on admission) [60], and occurs in 20–40% of ADHF patients, and has been traditionally associated with poor outcomes [61].

The EVEREST trial demonstrated that WRF during hospitalization correlates with changes in blood pressure, body weight, and natriuretic peptides [59]. The trial demonstrated that WRF accompanied by effective decongestion may not necessarily indicate worse outcomes.

The EURObservational Research Programme: Heart Failure Pilot Survey (ESC-HF Pilot) showed that chronic kidney disease (CKD) is a prevalent comorbidity in HF patients, and highlighted regional differences in patient characteristics and treatments, which may include variations in renal function and its management [62].

A Japanese study using the KUNIUMI registry, with 966 hospitalized ADHF patients, revealed that patients with both WRF and residual congestion at discharge had significantly worse outcomes compared to those without these conditions. The findings emphasize the importance of achieving adequate decongestion before discharge, particularly in elderly patients [63].

The evidence suggests that discharge decisions should not be based solely on renal function markers. Transient WRF in the context of successful decongestion should not necessarily delay discharge, while persistent WRF with residual congestion warrants extended hospitalization.

Moreover, recent research in HF has identified several important renal biomarkers that may facilitate earlier detection of kidney cellular damage in HF patients. These emerging renal markers include Kidney Injury Molecule-1 (KIM-1), Neutrophil Gelatinase-Associated Lipocalin (NGAL), and Interleukin-18 (IL-18). Each of these biomarkers offers unique insights into renal function and injury in the context of HF [64].

4.Hepatic markers

Hepatic congestion markers are valuable indicators for assessing HF severity and prognosis, particularly in ADHF.

The prevalence of abnormal liver function tests (LFTs) in HF patients is substantial, as listed in PROTECT study with baseline abnormalities observed in: aspartate transaminase (AST): 20%, alanine transaminase (ALT): 12%, Albumin: 40%, and predict worse outcomes; abnormal discharge LFTs had an unfavorable impact on 180-day mortality with hazard ratios (95% CI) for discharge AST: 1.5 (1.1–2.0), ALT: 1.5 (1.0–2.2), and albumin: 1.6 (1.2–2.1), respectively (all *p* < 0.05) [34].

Hepatic markers in HF can be broadly categorized into markers of necrosis/ischemia (ALT, AST, LDH (lactate dehydrogenase)), markers of stasis/congestion (total bilirubin (TBIL), ALP (alkaline phosphatase), GGT (gamma glutamyl transferase)), and markers reflecting both ischemia and congestion (albumin, INR (international normalized ratio)). In HF with low cardiac output, both ischemic and congestive markers may be elevated: acute ischemic injury characterized by sharp rises in ALT, AST, and LDH; chronic congestion reflected by elevated bilirubin, ALP, and potentially GGT; low albumin and prolonged prothrombin time may indicate the cumulative impact of both ischemia and congestion [65].

It is important to note that the pattern of liver dysfunction in HF can vary depending on whether the right or left side of the heart is predominantly affected. Right-sided HF tends to cause more hepatic congestion, while left-sided HF can lead to both congestion and ischemia.

However, an insight study from the TOPCAT trial proved that these markers alone were not consistently predictive of outcomes in HF with HFpEF [66].

Cholestasis markers TBIL and ALP also show strong prognostic value. Elevated TBIL correlates with increased risk of cardiovascular mortality, HF hospitalization, composite adverse outcomes [65], and also lower cardiac index (1.80 vs. 2.1; *p* < 0.001) and higher CVP (14.2 vs. 12.0; *p* = 0.03) [66].

Liver function abnormalities are dynamic markers that reflect hemodynamic status, congestion severity, and treatment response [67].

The integration of hepatic congestion markers into clinical assessment provides valuable prognostic information and helps guide treatment decisions in HF management. Regular monitoring of these parameters, particularly during hospitalization, can help optimize the timing of discharge and identify patients at higher risk for adverse outcomes.

5.Emerging biomarkers

Several emerging biomarkers show promise for use in ADHF, and we will note some of them below.

Carbohydrate Antigen 125 (CA125), traditionally known as an ovarian cancer marker, has gained attention in ADHF, serving as an indicator of congestion and predicting outcomes in AHF, with higher CA125 levels being associated with increased all-cause mortality (68% increase), higher rates of HF-related readmissions (77% increase), more severe fluid overload symptoms, and CA125 levels being significantly elevated in patients with pleural effusions, suggesting its utility in assessing fluid overload severity [68]. However, there are limitations of specificity, gender bias, and delayed response; CA125 levels may not change rapidly enough to guide acute management decisions.

Bioactive adrenomedullin (bio-ADM), a vascular-derived peptide hormone, has emerged as a promising biomarker for assessing congestion in ADHF, and it is produced in response to volume overload and acts to maintain endothelial barrier function. In ADHF patients, elevated bio-ADM levels correlate strongly with clinical signs of congestion, such as edema, orthopnea, and elevated jugular venous pressure. Studies have shown that bio-ADM tracks closely with mean right atrial pressure (mRAP) and is associated with measures of systemic congestion. A proposed cut-off value of 39 pg/mL has been identified for assessing congestion and predicting outcomes [69]. Higher pre-discharge bio-ADM levels reflect greater residual congestion, particularly peripheral edema, so that patients with elevated bio-ADM may benefit from more intensive diuretic therapy before discharge [70]. Regardless, there is limited clinical experience, and there is a lack of standardization, so more extensive clinical trials are needed to establish its role in routine practice.

Soluble suppression of tumorigenicity 2 (sST2), part of the IL-1 class involved in inflammatory signaling, is emerging as a valuable biomarker in ADHF, with elevated sST2 levels being correlated with vascular congestion, worse NYHA functional class, and increased 1-year cardiovascular mortality [71]. Nonetheless, sST2 can be elevated in various inflammatory conditions, optimal thresholds for risk stratification are not well-established, and there are limited data on treatment guidance.

Galectin-3 is a lectin molecule that promotes cardiac fibroblast activity, leading to left ventricular dysfunction. Like sST2, galectin-3 is recognized in guidelines for prognostication in chronic HF (Class IIb recommendation) [71]. However, its value in monitoring short-term treatment response is limited.

While these biomarkers show promise, it is important to note that many require further validation in large-scale studies before they can be routinely implemented in clinical practice. Their potential lies in complementing existing biomarkers to enhance the diagnosis, prognosis, and management of ADHF patients.

#### 2.1.3. Imaging Tools

A.Estimation of left ventricular (LV) filling pressures

Echocardiographic estimation of LV filling pressures (Figure 1) has become an indispensable tool in the management of HF patients (Appendix B, Table A3). The integration of multiple parameters, including left atrial volume indexed (LAVI), E/e’ ratio, and tricuspid regurgitation velocity, provides a more comprehensive assessment of congestion. Pre-discharge echocardiography, in particular, has demonstrated significant prognostic value and can optimize discharge timing and post-discharge care planning [72,73].

LAVI is a robust indicator of chronic elevation in LVFP (Figure 2). An enlarged left atrium, typically defined as LAVI > 34 mL/m^2^, reflects long-standing pressure overload [74]. The prognostic value of LAVI has been demonstrated in various studies. For instance, a meta-analysis by Shin et al. showed that increased LAVI was associated with adverse cardiovascular outcomes in patients with HFpEF [73]. Nonetheless, its clinical application faces several challenges. LAVI changes may not occur rapidly enough to guide acute management decisions in ADHF, and conditions such as atrial fibrillation and significant mitral valve disease can affect LAVI, complicating its interpretation in ADHF. These confounding factors necessitate careful clinical correlation and potentially limit LAVI’s standalone diagnostic value [74,75].

The ratio of early mitral inflow velocity (E) to mean septal and lateral early diastolic mitral annular velocity (e′) is widely used to estimate LVFP. An E/mean e′ ratio > 14 (E/e′ ≥ 11 for atrial fibrillation) suggests elevated filling pressures [72,77].

The Euro-Filling study, a multicenter investigation involving 159 patients, found that E/e′ was one of the most reliable echocardiographic parameters for estimating LV end-diastolic pressure (LVEDP) [78].

A tricuspid regurgitation velocity > 2.8 m/s indicates elevated right ventricular systolic pressure and possible right-sided congestion, this parameter being particularly useful when assessing pulmonary hypertension associated with left heart disease [72].

If other non-cardiac causes of pulmonary hypertension are excluded, an increase in sPAP (systolic pulmonary artery pressure) can be used as one of the criteria for the estimation of LVFP [43].

The optimal timing of echocardiography for guiding discharge decisions has been a subject of recent investigation. The OPTIMAL study provided relevant insights: admission echocardiography alone (within 3 days of admission) did not show a significant impact on survival (log-rank *p* = 0.33); pre-discharge echocardiography (performed within 3 days of discharge) was associated with significantly better survival compared to patients who did not undergo pre-discharge echo (log-rank *p* < 0.001). This survival benefit was observed across all HF phenotypes [79] and these findings highlight the importance of pre-discharge echocardiography in assessing treatment response, verifying hemodynamic stability, and guiding discharge timing.

Pre-discharge echocardiography may reveal residual congestion that is not apparent from clinical assessment alone. This information can guide further diuresis or other therapeutic interventions before discharge.

Echocardiographic parameters can help stratify patients according to their risk of readmission or adverse events post-discharge. For example, a study by Hoshida et al. in Osaka on 192 patients, showed that persistent increased diastolic elastance (Ed) to arterial elastance (Ea) ratio, as a relative index of LA pressure overload, at discharge was associated with higher rates of readmission in HFpEF patients [80].

Left atrial strain, measured by speckle tracking, has shown promise in estimating LVFP and detecting subclinical congestion. A study conducted by Cameli et al. showed that LA longitudinal deformation provided a better estimation of LVFP in a group of patients with LV ejection fraction lower than 35% [81].

As our understanding of echocardiographic techniques continues to evolve, future research should focus on standardizing protocols for congestion assessment and integrating these findings into clinical decision-support tools. By leveraging the full potential of echocardiography, clinicians can optimize the management of HF patients, and potentially reduce readmission and mortality rates.

B.The role of Lung ultrasound in congestion assesment

Lung ultrasound (LUS) has emerged as a valuable tool for assessing pulmonary congestion in HF, offering significant advantages over traditional methods. Its impact on congestion assessment and its role in discharge timing have been the subject of numerous studies in recent years.

LUS provides a more sensitive and specific method for detecting pulmonary congestion compared to conventional clinical examination and chest radiography. The technique implies that one should perform the LUS exam in a standardized manner to facilitate reproducibility, interpretation, and monitoring. The protocol implies a variable number of examination areas, varying from four zones to 28 zones. At the moment, the eight-zone examination is the most frequently used and the one recommended by the EACVI (European Association of Cardiovascular Imaging) [82].

LUS markers of congestion are B-lines and pleural effusion. While fluid in the pleural space represents a more severe marker of congestion, B-lines can be identified in the early stages of congestion and can guide rapid initiation of therapy. B-lines are easy to identify LUS artifacts; they appear as vertical, “laser-like” hyperechoic lines, starting from the pleural line and extending to the lower part of the interrogation sector, moving synchronously with the pleural sliding (Figure 1). Less than two B-lines per intercostal space can be identified in normal persons. However, an increasing number of B-lines will be present as interstitial lung edema appears [83]. The presence and quantity of B-lines on LUS correlate strongly with the degree of extravascular lung water and can accurately identify pulmonary edema [13].

A meta-analysis by Maw et al. demonstrated that LUS is significantly more sensitive than chest X-ray in detecting pulmonary edema in acute HF, with a relative sensitivity ratio of 1.2 (95% CI, 1.08–1.34; *p* < 0.001) [84]. This increased sensitivity allows for earlier detection of congestion, potentially leading to more timely interventions.

Palazzuoli et al. identified specific B-line thresholds associated with poor outcomes at discharge (the 28-zone scanning method): ≥22 B-lines for patients with HFrEF and ≥18 B-lines for those with HFpEF [85]. These findings suggest that LUS could be used to tailor discharge decisions based on HF phenotype.

Furthermore, LUS can detect subclinical congestion that may not be apparent on physical examination. Rivas-Lasarte et al. found that the presence of subclinical pulmonary congestion at discharge (≥5 B-lines on LUS in the absence of rales on auscultation) was a significant risk factor for rehospitalization, unexpected visits for HF worsening, or death at 6-month follow-up (hazard ratio 2.63; 95% confidence interval: 1.08–6.41; *p* = 0.033) [86].

A prospective study by Cohen et al. examined the association between B-lines on LUS at discharge and 30-day readmission risk in ADHF patients. They found that patients with 2–3 positive lung zones (≥3 B-lines per zone) had a 1.25 times higher risk of 30-day readmission (95% CI: 1.08–1.45), while those with 4–8 positive zones had a 1.50 times higher risk (95% CI: 1.23–1.82), compared to patients with 0–1 positive zones [87]. This study highlights the potential of LUS as a tool for risk stratification at discharge.

Moreover, LUS markers change rapidly during decongestion. A systematic review by Platz et al. reported that the B-line number changed as fast as 3 h after initiation of HF treatment. The other findings from this review support the previous statements, as it concludes that ≥15 B-lines on 28-zone LUS at discharge identified patients at a more than five-fold risk for HF readmission or death [88].

The BLUSHED-AHF trial, a multicenter, single-blind prospective clinical trial, showed that changes in LUS congestion scoring (LUS-CS) from emergency department admission to hospital discharge were associated with improved readmission-free survival [89]. Notably, the reduction in LUS-CS B. was most beneficial for heavily congested patients who had otherwise reassuring clinical features at admission. This suggests that LUS may be particularly useful in identifying patients who require more aggressive decongestion despite appearing clinically stable. Furthermore, LUS offers several advantages over traditional methods of assessing congestion, such as physical examination and chest X-rays, as it provides real-time, non-invasive, and quantifiable data on pulmonary congestion [90].

The DRY-OFF study further supported the utility of LUS by demonstrating its ability to track changes in pulmonary congestion throughout hospitalization, revealing different patterns of decongestion between HFrEF and HFpEF patients: HFrEF patients presented with higher indexes of pulmonary and intravascular congestion at admission compared to HFpEF patients. Also, at discharge, HFrEF patients still had higher B-line scores (0.4 ± 4 vs. 0.3 ± 0.4; *p* = 0.023), indicating more residual pulmonary congestion compared to HFpEF patients [91].

These findings underscore the potential of LUS as a valuable tool for tailoring decongestion strategies and optimizing treatment in ADHF patients during their hospital stay. Nonetheless, it has some limitations. The accuracy of B-line quantification can vary based on the operator’s experience and skill level. Different protocols, ranging from four-zone to 28-zone examinations, can lead to inconsistencies in results across studies and clinical practice. Factors such as obesity, subcutaneous emphysema, or pleural calcifications can affect image quality and B-line visualization. B-lines are not exclusive to ADHF and can be present in other lung pathologies like pneumonia, ARDS (adult respiratory distress syndrome), or pulmonary fibrosis. Both B-lines and BNP levels decrease with increasing BMI (body mass index) in HF patients, potentially limiting their ability to accurately predict congestion in obese individuals [92].

C.Estimation of right ventricular (RV) filling pressures

1.Inferior vena cava evaluation

Inferior vena cava (IVC) evaluation has emerged as a valuable tool for assessing congestion and guiding discharge decisions in patients with ADHF. This non-invasive ultrasound technique provides insights into right-sided filling pressures and overall volume status, potentially improving the management of HF patients.

IVC ultrasound assessment commonly involves measuring the maximum IVC diameter (IVCmax) and the collapsibility index (IVC-CI, formula in Appendix A) [93]. It is considered a normal IVC diameter when it is measured as <21 mm with collapsibility > 50%, while IVC is considered dilated when its diameter is >21 mm and collapsibility is <50% [94].

These parameters offer several advantages in congestion assessment. Studies have shown that these parameters are associated with clinical outcomes. A dilated IVC with reduced collapsibility suggests elevated right atrial pressure, indicating congestion and had greater risk of rehospitalization [95].

A study by Pellicori et al. demonstrated that increasing IVC diameter identifies patients with chronic HF at higher risk of adverse outcomes, regardless of LV ejection fraction [96]. A study by Cubo-Romano et al. found that a dilated IVC (>21 mm) at admission was associated with higher 90-day mortality after hospitalization for HF [97].

IVC measurements can be used to track changes in volume status over time, potentially guiding treatment decisions, the use of IVC evaluation in discharge decision-making being an area of active research. The CAVAL US-AHF trial, currently ongoing, aims to evaluate whether IVC and lung ultrasound-guided therapy can reduce subclinical congestion at discharge and improve outcomes in acute HF patients [93]. This study may provide valuable insights into the utility of IVC assessment in optimizing discharge timing. Preliminary results suggest that this approach may reduce subclinical congestion at discharge and lower the risk of readmission or death at 90 days.

While IVC evaluation shows promise, it is important to note that this procedure has its limitations. IVC measurements can be influenced by factors such as patient position, respiratory effort, mechanical ventilation, obesity, and operator technique. Moreover, rather than in isolation, IVC assessment should be used alongside clinical evaluation and other diagnostic tools [93,95].

2.Doppler Flow in the Hepatic Veins and the Portal veins

Doppler ultrasound evaluation of liver veins has emerged as a valuable tool for assessing congestion in ADHF and informing discharge decisions.

The hepatic vein Doppler waveform evaluation in healthy individuals shows a triphasic pattern, consisting of two antegrade flow peaks toward the heart and one retrograde flow peak toward the liver. This triphasic pattern reflects changes in blood flow through the right heart chambers during the cardiac cycle, with the waveform components corresponding to atrial contraction (A wave), ventricular systole (S wave), and ventricular diastole (D wave) [98].

This non-invasive technique provides crucial insights into the right heart hemodynamic status of patients, offering a more comprehensive assessment of congestion beyond traditional clinical parameters. In HF and congestion, the normal triphasic hepatic vein waveform undergoes characteristic changes that reflect the altered hemodynamics of the right heart. We can observe a mild augmentation of the A wave from the early stages of HF due to increased backflow from the failing right ventricle and spectral broadening and dampening of the retrograde pre-systolic wave. As congestion increases, the reversal of the S wave to D wave ratio can be observed, with the S wave becoming smaller than the D wave and the S wave may reach the baseline, indicating no antegrade flow during ventricular systole. During advanced stages (severe congestion), the triphasic pattern may be replaced by a biphasic or monophasic waveform. In severe cases, the A, S, and V waves may fuse into a single retrograde wave alternating with the D wave [99].

These hepatic venous waveform patterns, as determined by abdominal ultrasonography, have been shown to correlate with the degree of hepatic congestion and right atrial pressure [100]. A study by Landi et al. demonstrated that severe initial congestion was a strong predictor of inpatient mortality, HF-related death, and early readmissions [101].

Premature discharge with residual congestion can lead to the worst outcomes; therefore, the timing of discharge for ADHF patients is critical. Ultrasound evaluation of hepatic veins provides objective data that can guide this decision-making process. A study by Singh and Koratala highlighted that Doppler ultrasound assessment of hepatic and portal vein waveforms aids in monitoring the efficacy of decongestive therapy, potentially informing when a patient is ready for discharge [102].

Furthermore, research by Rivas et al. revealed that approximately 40% of patients have residual congestion that is only identifiable through lung ultrasound or abdominal vascular ultrasound [86]. This suggests that relying solely on clinical signs and symptoms may be insufficient for determining optimal discharge timing.

The prognostic significance of hepatic vein ultrasound findings extends beyond the immediate discharge period. A study published in the Journal of the American Heart Association introduced the hepatic venous stasis index (HVSI) as a quantitative measure of hepatic congestion (formula in Appendix A). The researchers found that a higher HVSI was associated with increased levels of BNP, larger IVC diameter, and higher mean right atrial pressure. Importantly, patients with higher HVSI values experienced more cardiac events during follow-up, indicating its potential as a prognostic marker [99].

While hepatic vein ultrasound is indeed valuable, its integration with other ultrasound parameters can provide a more comprehensive assessment of congestion. A study demonstrated that combining hepatic vein ultrasound according to the VExUS protocol with lung ultrasound had the greatest prognostic significance for predicting adverse outcomes in ADHF patients (HR = 16.7, 95% CI: 3.9–70.7; *p* < 0.001) [103].

Portal vein (PV) flow measurement represents an important component in evaluating congestion in ADHF, and it is obtained through Doppler ultrasonography, alongside hepatic vein flow patterns measurement. In normal conditions, PV flow is a continuous flow with minimal variations (pulsatility). However, in pathological states such as ADHF, increased right atrial pressure is transmitted retrogradely into the portal circulation, causing the flow to become pulsatile [104].

The portal vein pulsatility ratio (PVPR, formula in Appendix A) or portal vein pulsatility index (PVPI, formula in Appendix A) are commonly used to quantify this phenomenon, with PVPI < 30% being considered normal or associated with mild congestion, PVPI between 30% and 50% being associated with moderate congestion, while PVPI ≥ 50% is considered a marker of severe systemic congestion, and a decrease in PVPI of > 50% would indicate successful decongestion [105].

Kuwahara et al. found that on admission, PVPR was significantly higher in patients with acute HF compared to controls (0.29 ± 0.20 vs. 0.08 ± 0.07, *p* < 0.01). The same study observed a significant decrease in PVPR after improvement in HF (admission: 0.29 ± 0.20 vs. discharge: 0.18 ± 0.15, *p* < 0.01), suggesting its potential as a marker of decongestion [105].

PVPI on admission has been shown to inversely correlate with right ventricular function (tricuspid annular plane systolic excursion, ρ = −0.434) and positively correlate with pulmonary pressure (ρ = 0.346, *p* < 0.05). Bouabdallaoui et al. reported that 64% of ADHF patients on admission and 24% at discharge had abnormal PVPI (defined as ≥ 50%) [106].

PVPI assessment is often integrated with other ultrasound-based techniques for a comprehensive evaluation of congestion [104]. Combining PVPI with other metrics such as lung ultrasound and clinical parameters may provide a more accurate assessment of congestion and guide discharge decisions [107].

3.Ultrasonographic Evaluation of Intrarenal Venous Flow

Renal vein ultrasound has emerged as another valuable tool in the assessment of congestion in ADHF, complementing other ultrasound techniques such as hepatic vein and portal vein evaluation. This non-invasive method provides crucial insights into renal hemodynamics and systemic venous congestion and increased CVP, potentially influencing clinical decision-making and discharge timing.

Renal venous congestion is a key pathophysiological mechanism in ADHF that can lead to WRF and adverse outcomes [108]. Doppler ultrasound of the renal veins allows for direct visualization and quantification of venous flow patterns, offering a window into the hemodynamic status of patients.

The assessment typically involves Doppler evaluation of the intrarenal veins, with particular attention to the interlobar veins. In normal physiological conditions, renal venous flow is continuous with minimal phasicity. However, in the presence of increased CVP, as often seen in ADHF, the intrarenal venous flow (IRVF) pattern becomes increasingly phasic. This phasicity is categorized into different stages, reflecting the severity of congestion. It is considered mildly abnormal when the waveform is pulsatile, showing distinct systolic (S) and diastolic (D) components. This indicates some degree of venous congestion, but the flow is still maintained in both cardiac phases. When it is severely abnormal, the waveform becomes monophasic, displaying only a diastolic (D) pattern. This suggests significant venous congestion, where forward flow is only possible during diastole due to increased central venous pressure [109].

Several studies have characterized intrarenal venous flow patterns and their clinical significance in ADHF. As venous congestion worsens, IRVF progresses from continuous to biphasic and eventually monophasic patterns [110]. These patterns correlate with increasing right atrial pressures and severity of congestion.

A study by Iida et al. demonstrated that the IRVF pattern strongly correlates with right atrial pressure. They found that a discontinuous IRVF pattern was associated with a right atrial pressure ≥ 10 mmHg, with high sensitivity (90%) and specificity (86%) [111].

Beaubien-Souligny et al. conducted a prospective study on cardiac surgery patients and found that severe venous congestion, as indicated by Doppler anomalies in the portal and renal veins, was associated with a higher risk of acute kidney injury and longer duration of hospitalization [112]. This suggests that renal vein ultrasound could be a useful tool in identifying patients at risk of complications and prolonged hospital stays.

Moreover, emerging evidence suggests that renal venous US patterns may have utility in deciding the discharge timing. Improvement in renal venous Doppler parameters during hospitalization correlates with successful decongestion and better outcomes [113].

4.Ultrasonographic evaluation of the internal jugular vein

Ultrasonographic evaluation of the internal jugular vein (IJV) has emerged as a promising tool for congestion assessment in ADHF patients [114].

The IJV can be easily visualized using a high-frequency linear ultrasound probe placed over the neck in the area of the sternocleidomastoid muscle and then by moving the probe inferiorly to the angle of Louis [115]. The cross-sectional area of the IJV is measured at end-expiration and during the strain phase of a standardized Valsalva maneuver.

Simon et al. showed that a cross-sectional area (CSA) change of <66% in the IJV predicted right atrial pressure ≥ 12 mmHg with a positive predictive value of 87% [116]. This strong correlation between IJV ultrasound measurements and invasively measured right atrial pressures suggests that this non-invasive technique can reliably estimate CVP. Moreover, this study also demonstrated that normalization of IJV compliance (defined as CSA change ≥ 66%) at discharge had a 91% predictive value for avoiding 30-day readmission [116].

The ability of IJV ultrasound to predict readmission risk has significant implications for discharge timing. Pellicori et al. noted that a JVD ratio < 4 (maximal diameter during Valsalva divided by the diameter at rest) is abnormal and associated with severe symptoms, elevated natriuretic peptides, right ventricular dysfunction, and tricuspid regurgitation [115]. This suggests that achieving a normal JVD ratio could be a target for optimizing therapy before discharge.

A systematic review by Chaudhary et al. concluded that ultrasound interrogation of the IJV plays a significant role in determining the timing of discharge for patients admitted with ADHF [114]. The review emphasized the potential of IJV ultrasound parameters to guide clinical decision-making regarding readiness for discharge.

While ultrasound assessment of the IJV shows promise for guiding management in ADHF, several limitations should be considered. IJV ultrasound measurements can be affected by imaging technique, operator skill, and patient factors, potentially introducing inconsistencies; moreover, the IJV diameter varies with postural changes, which may affect measurements taken in different positions. The test relies on patients performing an adequate Valsalva maneuver, which may be challenging for severely ill individuals; also, some patients may have small anteroposterior IJV diameters for anatomical reasons, leading to low distensibility ratios even with normal right atrial pressure [117].

5.Chest Radiography

Chest X-ray (CXR) plays a significant role in the assessment of congestion in ADHF and, while having some limitations, remains a valuable component in the diagnostic process of ADHF patients.

Chest X-ray is widely used in the emergency department for patients presenting with acute dyspnea, serving as a first-line diagnostic imaging modality, being a fast and inexpensive procedure [118].

Pan, Pellicori et al. found that radiographic evidence of congestion was very common in ADHF patients and was associated with other clinical measures of worse prognosis: an increasing score based on the presence of pulmonary venous congestion, Kerley B-lines, pleural effusions, and alveolar edema was associated with all-cause mortality on multivariable analysis (hazard ratio 1.10, 95% confidence intervals 1.07–1.13, *p* < 0.001) [119].

Regarding discharge timing, the resolution of radiographic congestion can be an important factor in decision-making. However, it is crucial to note that CXR findings may lag behind clinical improvement. A study by Collins et al. demonstrated that serial CXRs are not recommended in the assessment of pulmonary congestion in chronic HF, as approximately one in five subjects with ADHF did not demonstrate signs of chest congestion on CXR [120].

While CXR remains a valuable tool, it has several limitations: radiation exposure, operator-dependent quality, interobserver variability, and detection of only severe extravascular pulmonary water variations. Due to these limitations, complementary tools are often used alongside CXR for a more comprehensive assessment of congestion. Lung ultrasonography (LUS) has emerged as a promising method to assess pulmonary congestion, offering advantages such as real-time evaluation and the ability to perform serial assessments without radiation exposure [121].

6.Bioimpedance Vector Analysis (BIVA)

Bioimpedance vector analysis (BIVA) offers a novel tool for assessing congestion in ADHF for both diagnosis and discharge timing.

This method represents a simple and quick clinical procedure involving placing four contact electrodes on the dorsal surface of the hand and foot. A low-intensity alternating current (~50 kHz) is passed through the body, and the resistance and reactance are measured. The resistance and reactance values, standardized for the patient’s height, are plotted on a Resistance-Reactance (R-Xc) graph. The resulting vector’s position, length, and angle provide information about the patient’s hydration status and body composition [122].

As for the detection of congestion in ADHF, BIVA has demonstrated superior accuracy compared to BNP in detecting peripheral congestion. A study by Massari et al. reported that the area under the curve (AUC) for BIVA was 0.88, significantly higher than the 0.57 for BNP, making algorithms that include BIVA together with NT-proBNP or BNP extremely accurate for patients’ prognosis assessment [123].

BIVA can identify subclinical congestion that may not be apparent through physical examination. This early detection capability allows for more timely interventions and potentially prevents decompensation [124]. Santarelli et al. demonstrated that when combined with clinical signs, BIVA showed a very good predictive value for cardiovascular events at 90 days (AUC 0.97) [125].

BIVA is considered an easy-to-use and low-cost method for assessing body composition and hydration status. It requires minimal patient preparation and can be performed quickly, making it suitable for routine clinical use [126]. However, it has several limitations: BIVA requires specialized equipment that may not be available in all healthcare settings; various protocols exist for BIVA assessment, which can lead to inconsistencies in results across studies and clinical practice; BIVA measurements can be affected by factors such as BMI, nutritional status, and the presence of pacemakers or implanted cardioverter defibrillators; moreover, the accuracy of BIVA measurements depends on the precision of the device used, which should be <1% for reliable evaluation of phase angle values and hydration status [127,128].

#### 2.1.4. Integrated Assessment Tools

Integrated assessment tools combine multiple parameters to provide a more comprehensive evaluation of congestion status. The timeline of integrated assessment tools has started simply with clinical congestion scores, with time adding the use of biomarkers and imaging.

Recent data from the RELAX-AHF-2 trial provide further evidence on the importance of achieving decongestion before discharge. Pagnesi et al. found that residual congestion at day 5 was present in 57% of patients and independently associated with worse outcomes: patients had more severe symptoms and comorbidities, received higher doses of loop diuretics but with lower response, were less likely to show hemoconcentration, and had higher rates of WRF. After adjusting for clinical variables, both any sign of residual congestion and a high clinical congestion score (CCS ≥ 3) at day 5 were independently associated with worse outcomes: any residual congestion: adjusted HR 1.32 (95% CI: 1.15–1.51, *p* < 0.001), CCS ≥ 3: adjusted HR 1.62 (95% CI: 1.39–1.88, *p* < 0.001) [129].

The RELAX-AHF-2 trial utilized a comprehensive congestion score that included assessment of orthopnea, pulmonary congestion, peripheral edema, and JVP. This scoring system demonstrated good predictive value for outcomes, supporting the use of multi-parameter approaches in evaluating congestion status [129].

ESC (European Society of Cardiology) viewpoint on congestion scores to use at pre-discharge:

According to the 2021 ESC guidelines [130], there is no single recommended congestion score to use at discharge for ADHF, but the ESC recommends an integrated approach to assess congestion before discharge, which includes:clinical evaluation—to assess symptoms and signs of congestion: dyspnea, orthopnea, peripheral edema, and jugular venous distension;hemodynamic parameters—to evaluate vital signs (blood pressure, heart rate), and to estimate glomerular filtration rate;biomarkers—to monitor NPs levels (BNP or NT-proBNP)—A reduction of ≥30% in NT-proBNP from admission to discharge being associated with better outcomes;imaging—to utilize echocardiography and lung ultrasound to assess for residual congestion.

The 2023 Focused Update of the 2021 ESC Guidelines for heart failure [131] emphasized the importance of close follow-up in the first 6 weeks after discharge, based upon the results of STRONG-HF trial [132], bringing attention to:symptoms and signs of congestion;blood pressure;heart rate;NT-proBNP values;potassium concentrations;estimated glomerular filtration rate.

While not explicitly recommending a specific congestion score, the ESC guidelines emphasize the importance of an integrative assessment of congestion status before discharge to reduce the risk of readmission and to improve outcomes.

##### Clinical Practice Scores

The EVEREST score is most commonly used in practice and it was developed by Ambrosy et al. in 2013 within the EVEREST (Efficacy of Vasopressin Antagonism in Heart Failure Outcome Study with Tolvaptan) trial, and it includes assessment on a standardized scale ranging from 0 to 3 of orthopnea, dyspnea, rales, JVD, peripheral (pedal) edema, fatigue at discharge [5]. Each component is scored between 0 to 3, based on severity; higher scores indicate more severe congestion, the highest score being 18. An EVEREST score < 2 has been proposed as a decongestion target for patients admitted with ADHF [106].

The Venous Excess Ultrasound (VExUS) score has emerged as a non-invasive point-of-care ultrasound (POCUS) approach to evaluate systemic venous congestion, facilitating a comprehensive evaluation of venous congestion and guiding therapeutic management.

The VExUS score is calculated in a stepwise manner, beginning with the measurement of IVC diameter, followed by Doppler waveform analysis of intra-abdominal veins if the IVC suggests congestion (Figure 3) [109].

IVC Diameter Assessment—The assessment begins with measuring the IVC diameter at its widest point during quiet respiration in the subcostal view. An IVC diameter ≥ 20 mm indicates elevated CVP, which is often associated with systemic venous congestion. This finding necessitates further evaluation of venous Doppler waveforms to confirm and grade the severity of congestion;Hepatic Vein Doppler Waveform Analysis—The hepatic vein Doppler waveform typically exhibits triphasic flow, with a dominant systolic (S) wave under normal conditions. As venous congestion worsens, the S wave diminishes relative to the diastolic (D) wave, and in severe cases, S wave reversal occurs;Portal Vein Pulsatility Index (PVPI)—The portal vein normally demonstrates continuous flow with minimal pulsatility. It increases with elevated CVP due to direct transmission of right atrial pressure, and a PVPI ≥ 50% indicates severe portal venous congestion;Renal Vein Doppler Waveform Analysis—The renal vein Doppler waveform is normally continuous and monophasic. With increasing venous congestion, the waveform becomes discontinuous or biphasic, and, in severe cases, it transitions to monophasic diastolic flow only;Grading system: The final VExUS grade is assigned based on the combination of IVC diameter and the severity of Doppler abnormalities in the hepatic, portal, and renal veins:
Grade 0: IVC diameter < 20 mm, with no Doppler abnormalities, indicating no congestion;Grade 1: IVC diameter ≥ 20 mm with any combination of normal or mildly abnormal venous Doppler patterns;Grade 2: IVC diameter ≥ 20 mm with one severely abnormal venous Doppler pattern (S wave reversal in the hepatic vein, PVPI ≥ 50% in the portal vein, or discontinuous monophasic flow in the renal vein);Grade 3: IVC diameter ≥ 20 mm with two or more severely abnormal venous Doppler patterns.


The VExUS score at discharge has shown promise in identifying patients at higher risk for readmission or adverse events [107].

Rinaldi et al. conducted a prospective cohort study that found patients with a VExUS score of 2 or 3 at discharge had a higher risk of readmission or emergency visits due to ADHF within 90 days compared to those with lower scores [107].

Collins et al. (2025) highlight the potential of the VExUS score, which combines multiple ultrasound parameters to evaluate systemic venous congestion. This integrated approach shows promise in predicting readmission risk at discharge, potentially offering a more comprehensive assessment of congestion status [133].

##### Integrative, Multi-Parameter Approaches

Looking at the timeline of patient assessment research, there is a standardized vision of an integrative assessment method for ADHF, in both admission and pre-discharge management, but what differentiates the studies mainly is how comprehensive and detailed they are, and how they include more specific biomarkers and/or imaging techniques.

As for the research from the past ten years, which tended to focus more on the general concept of integrative assessment, the latest years provided a more structured and detailed framework. The combination of multiple methods to improve accuracy with clinical assessment, biomarkers with their representative NPs, imaging with echocardiography and lung US, has been discussed in the past ten years [4].

As noted in a recent study published by Lavalle et al., which highlights the efficacy of new HF therapies in specific patient subgroups, it is important to fully integrate new HFrEF therapies also in specific subgroups of patients, demonstrating yet again the need for their full implementation in clinical practice, based on comprehensive assessments [134].

Another recent study, conducted by Collins et al. (2025), underscores the importance of timely discharge decisions based on comprehensive assessments. They stress that premature discharge with residual congestion can lead to poor outcomes, emphasizing the need to thoroughly evaluate congestion status before finalizing discharge plans [133].

As we discover newer biomarkers and newer imaging modalities, there is a shift towards the implementation of newer integrative assessment methods that include hemoconcentration markers and also emerging ones that complement existing biomarkers, venous US with VExUS score, and even telemedicine and remote monitoring for early detection [2].

Telemedicine enables continuous monitoring of various parameters that can indicate worsening HF. Patients can transmit daily measurements of blood pressure, heart rate, and oxygen saturation and can report subjective symptoms like dyspnea, fatigue, and edema through questionnaires [135]. A novel approach to telemedicine in ADHF management involves a patient-performed lung ultrasound. A pilot study conducted by Chiem et al. demonstrated the feasibility of training HF patients to perform LUS self-exams at home [136]. Patients were trained using a 15-min tutorial video to perform a four-zone LUS using a handheld ultrasound device. An amount of 85% of lung zones examined by patients were interpretable by expert reviewers, with high agreement between experts. This method has a very high rate of patient acceptance, with 98% of participants reporting they could perform the LUS self-exam at home [136].

Wearable technology offers promising opportunities for enhancing ADHF management by providing real-time data and facilitating personalized care, by monitoring vital signs, detecting early signs of decompensation, and ultimately improving patient outcomes by enabling timely interventions and reducing hospital readmissions. Further research is needed to fully leverage these technologies, with data accuracy and integration into clinical workflows being notable challenges [137].

As the new era of artificial intelligence (AI) and machine learning (ML) is evolving at an exponential rate, the ADHF management landscape is also expanding on their account. These technologies can help identify patterns that may not be apparent through traditional assessment methods, potentially leading to more accurate predictions of patient outcomes and personalized treatment strategies. However, one should keep in mind that there still exist limitations of current AI models in real-world scenarios and their application has remained constrained by ongoing challenges [138].

##### Proposed Algorithm for Pre-Discharge Assessment

Residual congestion at discharge is a risk factor for hospital readmissions and poor outcomes; therefore, assessment and optimization of fluid balance at end-hospitalization should be standard of care. Several clinical, paraclinical, and imaging biomarkers of congestion have been proposed and a multivariable assessment is advised in the light of the current scientific literature data. Clinical and US biomarkers are readily available and non-expensive tools that can be repeated throughout hospitalization. Laboratory parameters, however, can be at times expensive, and their repeated usage is limited by financial barriers such as in the case of NPs. Nonetheless, hemoconcentration or hepatic parameters can be used without much added cost. Ideally, an integrative approach with clinical, imaging, and biomarker analysis should be used to plan hospital discharge and to ensure complete decongestion in order to improve HF patients’ outcomes (Figure 4). At the moment, the optimal “toolbox” to use in HF patients at discharge is still to be defined; however, it can be assumed that integrating individual tools with proven benefits in the management of HF patients at end-hospitalization and post-discharge period will improve outcomes in this HF population.

## 3. Conclusions

Successful decongestion prior to discharge is crucial for patients with ADHF to reduce the risk of readmission and improve outcomes. Residual clinical congestion at discharge is a strong predictor of mortality and readmissions. Unfortunately, the optimal assessment of decongestion and how to guide treatment have not been clearly defined nor have they been standardized. We here propose an integrative approach for a comprehensive clinical evaluation involving the assessment of symptoms, signs, biomarkers, and imaging, which is essential in determining readiness for discharge.

## Figures and Tables

**Figure 1 medicina-61-00816-f001:**
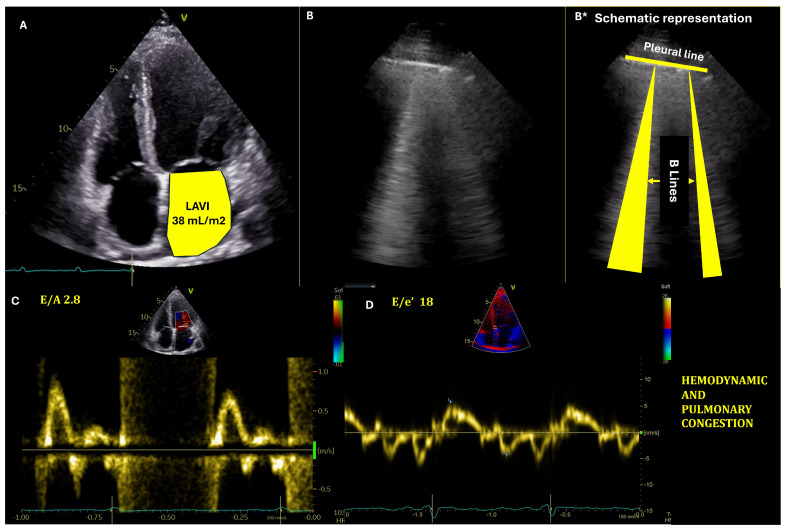
Assessment of hemodynamic and pulmonary congestion with echocardiography and lung ultrasound. (**A**) Transthoracic echocardiography, four-chamber view depicting dilated left atrium. (**B**,**B***): Lung ultrasound with lung congestion—presence of B-lines (laser-like artefacts beginning at the pleural line and ending at the lower part of the interrogation sector). (**C**,**D**): Pulsed wave Doppler at the mitral valve and tissue Doppler tracing at the interventricular septum, respectively, showing increased left ventricular filling pressures.

**Figure 2 medicina-61-00816-f002:**
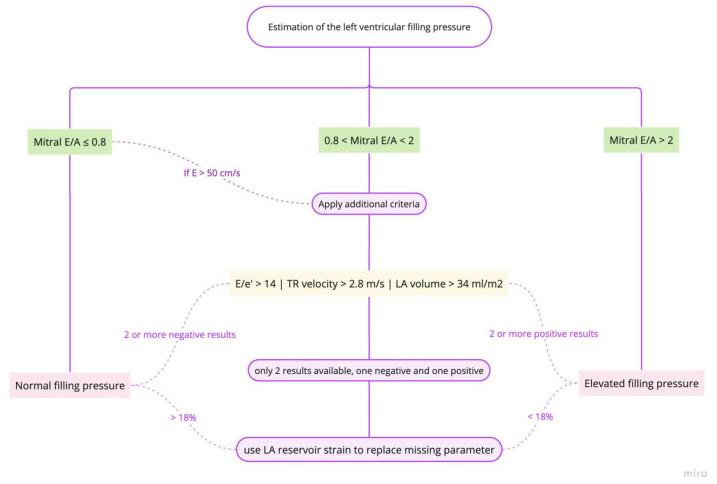
Algorithm for estimation of left ventricular filling pressure (LVFP) (modified from [76]). A = the late filling wave due to atrial contraction, E = the early rapid filling wave velocity, e′ = mitral annular early diastolic velocity, CRT = cardiac resynchronization therapy, HF = heart failure, LA = left atrium, LBBB = left bundle branch block, LV = left ventricular, MAC = mitral annular calcification, MR = mitral regurgitation, MS = mitral stenosis, MV = mitral valve, RV = right ventricular, TR = tricuspid regurgitation.

**Figure 3 medicina-61-00816-f003:**
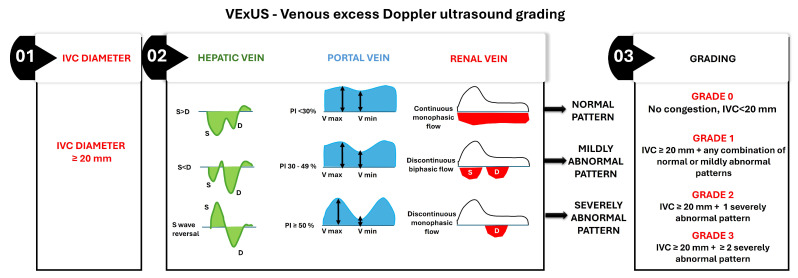
VExUS (venous excess Doppler ultrasound) grading score—step by step approach. IVC—inferior vena cava, PI—pulsatility index.

**Figure 4 medicina-61-00816-f004:**
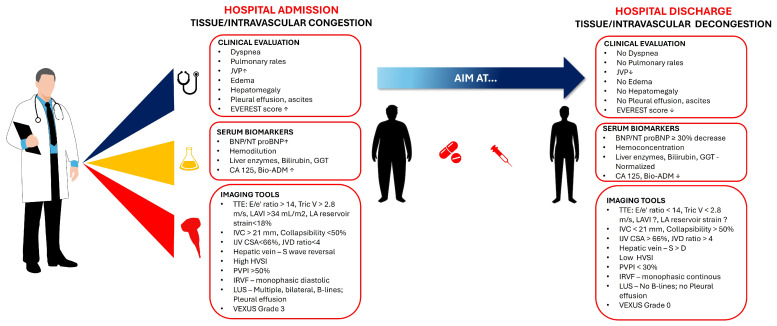
Proposed integrated algorithm for admission and pre-discharge assessment. At discharge, an “ideal” decongestive status is described. BNP—B-type natriuretic peptide, GGT—gamma glutamyl transferase, HVSI—hepatic venous stasis index, IJV CSA—internal jugular vein cross sectional area, IRVF—intrarenal venous flow, JVD—Jugular vein diameter, JVP—jugular venous pressure, LAVI—left atrium volume indexed, LUS—lung ultrasound, NT proBNP—N terminal proBNP, PVPI—portal vein pulsatility index, TTE—transthoracic echocardiography, Tric V—tricuspid velocity, VExUS—venous excess ultrasound score, ↑—increased.

## Data Availability

The data presented in this study are available upon request from the corresponding author.

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
