# Peer review of "Assessment of Decongestion Status Before Discharge in Acute Decompensated Heart Failure: A Review of Clinical, Biochemical, and Imaging Tools and Their Impact on Management Decisions"

_medicina, 2025, doi:10.3390/medicina61050816_

Round 1

Reviewer 1 Report

Comments and Suggestions for Authors

In my opinion, this article «Assessment of Decongestion Status Before Discharge in Acute Decompensated Heart Failure: A Review of Clinical, Biochemical, and Imaging Tools and Their Impact on Management Decisions» is of great scientific interest. The authors have done a great job, summed up and analyzed a large amount of data. The only one point that I would advise the authors to pay attention to is the optimization of keywords.

Author Response

Dear Reviewer, 

We would like to express our sincere appreciation for the time and expertise you dedicated to reviewing our manuscript. 

We are grateful for the feedback you provided on our manuscript. We optimized the keywords, and we attached the revised manuscript. 

Thank you again for your contributions to our research.

Best regards,

Reviewer 2 Report

Comments and Suggestions for Authors

Congratulations to the authors for having written such an interesting review; here you can read my comments about:

The application of artificial intelligence and machine learning in multimodal assessment tools is a developing area that warrants brief discussion. Additionally, wearable health devices and telemonitoring technologies play an increasing role in congestion evaluation.

Including a comparative table outlining the sensitivity, specificity, and usability of various congestion assessment tools would enhance clarity. Moreover authors should include and briefly discuss the latest evidences in HF drug thrapy (doi: 10.1159/000541393)

A flowchart illustrating the proposed discharge assessment process would provide a practical visual guide for implementation.

Author Response

Dear Reviewer, 

Thank you for the thoughtful feedback on our manuscript. We have carefully considered each of your suggestions and made adjustments to improve the quality of our work as suggested. 

Below, we outline how we have addressed each of your comments:

  1. “The application of artificial intelligence and machine learning in multimodal assessment tools is a developing area that warrants brief discussion“ : we have added information in subsection 2.1.4, from row 1003 to row 1009, with supporting reference number 140. 
  2.  “Additionally, wearable health devices and telemonitoring technologies play an increasing role in congestion evaluation” : we have added information in subsection 2.1.4, from row 997 to row 1002, with supporting reference number 139. 
  3. “Including a comparative table outlining the sensitivity, specificity, and usability of various congestion assessment tools would enhance clarity“ : we have created and included tables in Appendix B so that they would not burden the rest of the article, but we provided citations in the main text for each of them, as we included a table for the clinical signs, a table for biochemical tools, and a table for imaging tools, each containing their description, clinical relevance, diagnostic utility, and prognostic value. Citations in the main text: row 104, row 116, row 260 and row 488. 
  4. “Moreover authors should include and briefly discuss the latest evidences in HF drug thrapy (doi: 10.1159/000541393)” : we have added information about the latest evidence in HF drug therapies and referenced the provided article to support the data.  (Rows 972-976) However, as the article focuses on the available tools for congestion assessment, we would like to not add further text regarding treatment as it would lengthen the article and broaden too much the area of discussion.
  5. “A flowchart illustrating the proposed discharge assessment process would provide a practical visual guide for implementation” : We agree with the Reviewer that flowcharts are useful to inform the readers this is why we have proposed figure 4 as a visual tool that reflects the data provided in the article (proposed integrated algorithm for admission and pre-discharge assessment). We think that adding a fifth figure would be redundant and lengthen the article too much. Nonetheless, we will change figure 4 if the Review has suggestions regarding it.

The revised manuscript is attached. We hope these changes meet your expectations and improve the overall quality of our submission.

We appreciate your time and effort in helping us enhance our submission, and look forward to your further assessment.

Thank you again for your contributions to our research.

Round 2

Reviewer 2 Report

Comments and Suggestions for Authors

Congratulations to the authors for the revised version of the manuscript.